# Physical activity in former elite cricketers and strategies for promoting physical activity after retirement from cricket: a qualitative study

Stephanie R Filbay,[1] Felicity L Bishop,[2] Nicholas Peirce,[3,4] Mary E Jones,[5] Nigel K Arden[1]

[1]Arthritis Research UK Centre for Sport, Exercise & Osteoarthritis, Nuffield Department of Orthopaedics, Rheumatology & Musculoskeletal Sciences, Botnar Research Centre, University of Oxford, Oxford, UK
[2]Department of Psychology, University of Southampton, Southampton, UK
[3]National Centre For Sports and Exercise Medicine and National Cricket Performance Centre, Loughborough University, Loughborough, UK
[5]Nuffield Department of Orthopaedics, Rheumatology & Musculoskeletal Sciences, University of Oxford, Oxford, UK
[4]National Cricket Performance Centre, Loughborough University, Loughborough, UK

**Correspondence to**
Dr Stephanie R Filbay;
stephanie.filbay@uq.net.au

## ABSTRACT

**Objectives** The health benefits of professional sport dissipate after retirement unless an active lifestyle is adopted, yet reasons for adopting an active or inactive lifestyle after retirement from sport are poorly understood. Elite cricket is all-encompassing, requiring a high volume of activity and unique physical demands. We aimed to identify influences on physical activity behaviours in active and insufficiently active former elite cricketers and provide practical strategies for promoting physical activity after cricket retirement.

**Design** 18 audio-recorded semistructured telephone interviews were performed. An inductive thematic approach was used and coding was iterative and data-driven facilitated by NVivo software. Themes were compared between sufficiently active and insufficiently active participants.

**Setting** All participants formerly played professional cricket in the UK.

**Participants** Participants were male, mean age 57±11 (range 34–77) years, participated in professional cricket for 12±7 seasons and retired on average 23±9 years previously. Ten participants (56%) were classified as sufficiently active according to the UK Physical Activity Guidelines (moderate-intensity activity ≥150 min per week or vigorous-intensity activity ≥75 min per week). Eight participants did not meet these guidelines and were classified as insufficiently active.

**Results** Key physical activity influences were time constraints, habit formation, intrinsic and extrinsic motivation, physical activity preferences, pain/physical impairment and cricket coaching. Recommendations for optimising physical activity across the lifespan after cricket retirement included; prioritise physical activity, establish a physical activity plan prior to cricket retirement and don't take a break from physical activity, evaluate sources of physical activity motivation and incorporate into a physical activity plan, find multiple forms of satisfying physical activity that can be adapted to accommodate fluctuations in physical capabilities across the lifespan and coach cricket.

**Conclusions** Physically active and less active retired cricketers shared contrasting attributes that informed recommendations for promoting a sustainable, physically active lifestyle after retirement from professional cricket.

### Strengths and limitations of this study

► A purposive sampling strategy was used to capture contrasting physical activity behaviours and experiences, enabling comparisons between sufficiently active and insufficiently active individuals.
► The study may have been subjected to selection bias, individuals who desire participation in a qualitative interview may differ from those who decline participation.
► The interviewer was a physiotherapist with knowledge of cricket and sports medicine and experience in interviewing and building rapport with individuals. Strong rapport enabled participants to share personal perspectives in a reflective and open manner that enriched the findings of this study.

## INTRODUCTION

When an individual adopts an inactive lifestyle, maladaptive responses lead to metabolic dysfunction increasing the risk of developing chronic disease.[1] Physical inactivity increases the relative risk of stroke by 60%, coronary artery disease by 45%, hypertension by 30% and diabetes by 50%, resulting in profound personal, societal and economic costs.[2] In contrast, regular sport participation is associated with a wide array of psychological, social and physical health benefits.[3 4] However, the physical benefit of sports participation dissipates following sport cessation; elite athletes who become inactive after retirement from sport face the same, or worse, risk of developing chronic disease as the inactive general population.[5] If the physical and psychological benefits an athlete obtained through professional sport could be maintained by adopting a physically active lifestyle after retirement, a career in professional sport could pave the way for a fulfilling and active life with multiple health benefits. In order to develop

strategies for promoting physical activity after retirement from sport, a greater understanding of reasons for physical inactivity in this population is needed.

Cricket is a popular team sport played by people of all ages across various continents. A professional cricketer must dedicate a large proportion of daily life to being physically active as games are often played over entire days and can last up to five consecutive days in duration. During the course of the 7-month summer season, the playing schedule is relentless and many elite cricketers also play overseas during the winter period. Elite cricketers train during the preseason months and in between games with a mixture of skills practice, aerobic and strength-based conditioning. Individuals who become professional cricketers have typically been training and playing large volumes of cricket since childhood, making cricket an ideal sport to explore physical activity behaviours after retirement.

The vast majority of cricket-related research has focused on incidence, prevention, prediction and treatment of cricket injuries.[6–13] A professional cricket career predisposes players to injury[6 8 14–17] which also places a professional cricketer at risk of developing osteoarthritis in later life.[18–22] Developing symptomatic osteoarthritis after retirement from cricket has potential to negatively impact physical activity levels in former cricketers, although this has not yet been explored. The Professional Cricketers' Association (PCA) published an online report from a past player survey of 506 former cricketers of mean age 49 (range 22–86) years.[23] The PCA reported that 88% of former cricketers needed to find work after retiring from cricket and 20% suffered health consequences from playing sport.[23] Transitioning from professional sport to a sedentary profession and health consequences from playing sport have potential to impact physical activity behaviours in former cricketers, although this was not investigated by the PCA. A better understanding of the factors that influence physical activity levels in retired professional cricketers will enable the design of interventions and strategies to support cricketers to adopt a sustainable physically active lifestyle after cricket retirement. Such insights may also be applicable to other professional athletes.

The aim of this study was to draw on retired cricketers' personal perspectives and experiences to

i. identify key influences on physical activity behaviours after retirement from professional cricket in sufficiently active and insufficiently active individuals
ii. provide practical strategies for promoting a physically active lifestyle after retirement from professional cricket.

## METHODS

This study is reported in accordance with the consolidated criteria for reporting qualitative research guidelines.[24]

### Recruitment

Participants were purposively sampled from a cohort of 187 former elite English cricketers. The cohort had been recruited from the former player membership list maintained by the PCA as part of a cross-sectional retrospective questionnaire study. The questionnaire collected information regarding cricket playing history, injury history, current joint health, medical history and demographics.[25] From this larger cohort, 143 participants indicated a willingness to be contacted again and formed the cohort from which participants were invited to the current study. Responses from two items in this questionnaire were used to allocate participants to one of two groups for purposive sampling: (i) individuals who strongly agree or agree that participation in cricket has resulted in an *increase* in current physical activity level (n=46, 42%) or (ii) Individuals who strongly agree or agree that participation in cricket has resulted in a *decrease* in current physical activity levels (n=27, 25%). This sampling strategy was used to capture contrasting physical activity behaviours and experiences to enable comparisons between sufficiently active and insufficiently active individuals. When selecting former cricketers to invite into the study, potential participants were purposely selected to ensure the sample represented men of varying ages. When an individual declined the invitation, a former cricketer of a similar age was invited into the study. Invitations and study information (including study rationale, procedure, dissemination plans and the interviewer's credentials) were sent via email. In total, 42 invitations were sent to eligible participants, 19 received no response, 2 people declined to participate, 2 people were unavailable due to overseas travel and 1 person did not respond to further correspondence despite an initial desire to participate. If no reply was received within 2 weeks, a new individual was invited into the study.

### Interviews

Informed verbal consent was obtained from each of the 18 participants prior to performing audio recorded semi-structured telephone interviews (mean duration 26 min (range 18–37 min)). All interviews were performed by SRF, a female physiotherapist and postdoctoral researcher with qualitative research experience who had not met the participants prior to interview. Interviews were transcribed verbatim by a research assistant, an alias was allocated to each participant and transcripts were deidentified during transcription. The semistructured interview guide was pilot tested with three people with cricket experience prior to ethics approval. This resulted in the addition of three questions (Q2, Q4, Q15) and the modification of one question (Q17) to assess the perceived relationship between current quality of life and an individual's past cricket career (box). The interview guide addressed key areas of interest while allowing the researcher to adapt the interview guide to elicit relevant and rich information from respondents through probing and prompting.[26] Open-ended questions provided participants with the

## Box    Semistructured interview guide

1. Can you describe any physical activity, exercise or sport that you currently take part in?
2. Has that remained fairly constant since you retired from cricket or has it changed over the years?
3. Have you played cricket again since retiring? Why/why not?
4. What was your motivation for playing cricket?
5. Are you as physically active as you would like to be? If no, why not? How does this make you feel?
6. What is your motivation for taking part in physical activity/exercise/sport?
7. How important is being physically active to you? (If important, why is it important?/If not important, has it always been this way?)
8. Does the type of physical activity that you do matter to you, or would you be satisfied taking part in any form of physical activity? (ask about specific forms of exercise that they find *dissatisfying* and why)
9. What physical activity goals are you currently trying to achieve, if any?
10. What are the barriers or challenges, if any, that impact on your ability to be physically active?
11. Do you think that retired cricketers face the same challenges with being physically active as the general population, or are they unique or different in some way?
12. Some retired cricketers become physically inactive, what advice would you give to help them maintain a physically active lifestyle after retiring from cricket?
13. If you wanted to increase your physical activity levels, what do you think would help you to do so?
14. Can you describe any positive or negative impacts that your previous participation in cricket has had on your current physical activity patterns?
15. If you hadn't played professional cricket, do you think that you would be more or less active than you currently are?
16. Does your current ability to participate in physical activity impact on your quality of life? If yes, in what ways? If no, why not?
17. Overall how satisfied are you with your current quality of life? Do you think that this is related to your past career in cricket?
18. Is there anything more you would like to add about your experiences with physical activity after retiring from professional cricket?

opportunity to consider personal perspectives and experiences (box). The interview guide was iteratively adapted throughout the interviews to incorporate any additional issues of importance to respondents (eg, by adding a question to explore their relationship with cricket post retirement). Participants had the opportunity to contribute any additional information at the end of the interview.

Data saturation was achieved by the 14th interview, defined a priori as the point at which no new themes were identified from four consecutive interviews (two from participants with increased physical activity and two from participants with decreased physical activity). Once data saturation was reached, an additional four interviews were performed to expand on ideas and themes after following the semistructured interview guide. If these final interviews resulted in the identification of new themes, additional interviews were planned until data saturation was again satisfied. No new themes emerged from these additional four interviews affirming data saturation. Data from all 18 interviews were used for analysis.

### Analysis procedure

The analysis procedure is summarised in figure 1. An inductive thematic approach was used[27 28] facilitated by NVivo V.11 software.[29] A study journal was used to summarise each interview and reflect on initial ideas. Transcripts were read multiple times with accompanying audio to identify all information potentially relevant to the research aims.[30] This information was coded into multiple categories to be later refined and analysed for themes.[27] Data coding was iterative and data-driven, performed without engagement with literature to avoid sensitisation to themes and without reference to a pre-existing coding structure.[27 30]

During subsequent stages of analysis, the data were further analysed for repeated patterns, codes were sorted into a hierarchical structure representing themes and subthemes, overlapping themes were merged and those outside the scope of the current study were filed separately. These themes and subthemes were repeatedly reviewed and refined to confirm external heterogeneity and internal homogeneity within themes and to ensure an accurate representation of the entire dataset. The study journal was also revisited to check that themes accurately reflected the key issues discussed by participants.[27 31] Themes were compared among sufficiently active and insufficiently active participants to better understand factors influencing physical activity behaviours.

A selection of six transcripts representing participants with diverse physical activity patterns were analysed by a

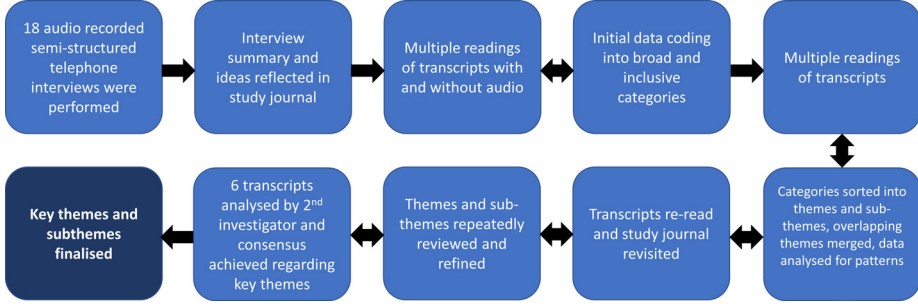

**Figure 1** Data analysis process.

second investigator (FLB) blinded to the coding structure developed by the first author (SRF). A meeting was then held between investigators and agreement was achieved regarding key themes in relation to these transcripts. Although no modifications were made to the coding structure following this meeting, the second investigator contributed to the consolidation and interpretation of key themes. Key themes and strategies for promoting physical activity will be described with reference to participant quotes[27 31] and in relation to relevant participant characteristics (ie, physical activity level, satisfaction/dissatisfaction with activity level and the presence/absence of joint pain).

### Physical activity classification

To enable comparison of physical activity behaviours and perspectives in active and less active counterparts, participant descriptions of current activity level over a typical week were used to categorise participants into 'sufficiently active' (meeting the UK Physical Activity Guidelines[32]) and 'insufficiently active' (not meeting the UK Physical Activity Guidelines[32]) groups. Participants were asked to describe any 'physical activity, exercise or sport' that they currently take part in and were prompted to provide details regarding activity type, duration, intensity and frequency and to assure responses reflected a typical week. The UK Physical Activity Guidelines recommend adults undertake moderate-intensity activity at least 150 min per week or vigorous-intensity activity at least 75 min per week for health-enhancing benefits including reduced chronic disease susceptibility and burden.[32] Physical activity type was categorised into moderate or vigorous intensity with reference to previous recommendations in accordance with the Centers for Disease Control and American College of Sports Medicine guidelines.[33]

### Participant characteristics

Participants were all male, aged a mean 57±11 (range 34–77) years and had been retired from professional cricket for an average 23±9 (range 7–38) years. Ten participants (56%) were sufficiently active, meeting or exceeding the UK Physical Activity Guidelines and eight participants (44%) were insufficiently active to meet these guidelines. One in two (n=9, 50%) would prefer to be participating in a greater volume of physical activity. Ten participants (56%) reported having received a diagnosis of osteoarthritis and 15 participants (83%) experienced joint pain (n=6, 40% of participants with joint pain had not been diagnosed with osteoarthritis). Full participant characteristics are presented in table 1.

### RESULTS
### Key influences on physical activity behaviours after retirement from professional cricket
#### Time constraints

The most common physical activity barrier identified by retired cricketers who expressed that they would like to be more active was time constraints. Many participants were working long hours in sedentary professions which was a stark contrast from life as a professional cricketer and resulted in difficulty finding the time to be physically active.

> Cam: 'Work takes up too much time, office based. I don't necessarily get as much time as I'd like either before, during or after work to, you know, do some physical activity, other stuff has to take priority.' *(51–55 years, insufficiently active, dissatisfied with activity level, current joint pain)*

> Lee: 'It's time, you know, I came out of cricket, in my 30's and you try and find your way and then you try set up a business and that sort of takes over really, so some days you just don't, you don't get chance to go out there and do things so readily.' *(46–50 years, sufficiently active, dissatisfied with activity level, current joint pain)*

In contrast, participants who were sufficiently active and satisfied with their physical activity levels prioritised physical activity, and irrespective of work and family commitments, allocated time to be physically active on a daily basis.

> Dan: 'You know time is limited and you have to vacate your time appropriately, but as long as you can build that into your regular routine then it doesn't tend to be so much of a problem.' *(56–60 years, sufficiently active, satisfied with activity level, current joint pain)*

> Interviewer: 'Have you ever struggled with regards to having enough time to exercise?'

> Joe: 'No, always make time.' *(61–65 years, sufficiently active, satisfied with activity level, no joint pain)*

### Habit formation

Retired cricketers not meeting the physical activity guidelines who were dissatisfied with their current physical activity level had difficulty establishing an exercise routine and integrating regular physical activity into their daily life. These individuals described adopting 'poor habits' early after retirement that were difficult to break when physical activity desires changed.

> Fin: 'Part of it I think it's habit and routine to be honest. Because saying I haven't got time for it is a lame excuse, because a lot of people work full time. Part of it is I've just got into such a bad habit and it's just mentally getting back into that, into sort of the boredom of physical activity … So I enjoyed having the break, but then obviously following on from that I never really turned it back around. So it was a choice to start with but then but it was a bad choice because it then meant that I didn't do anything … I then found it hard to find any kind of routine where it meant I actually went to the gym or did some activities.' *(31–35 years, insufficiently active, dissatisfied with activity level, current joint pain)*

**Table 1** Participant characteristics

| Alias | Age range* | Years post retirement* | UK professional seasons* | Body mass index | Joint pain | Osteoarthritis | Total joint replacement | Meeting physical activity guidelines | Are you as active as you would like to be? |
|-------|-----------|----------------------|-------------------------|-----------------|-----------|----------------|------------------------|--------------------------------------|-------------------------------------------|
| Dan | 56–60 | 26–30 | 6–10 | Normal | Yes | Yes | No | Yes | Yes |
| Dom | 61–65 | 26–30 | 16–20 | Obese | No | Yes | Yes | Yes | Yes |
| Gus | 56–60 | 11–15 | 1–5 | Overweight | No | No | No | Yes | No |
| Guy | 46–50 | 21–25 | 1–5 | Obese | Yes | No | No | Yes | No |
| Jim | 66–70 | 21–25 | 21–25 | Overweight | Yes | Yes | Yes | Yes | Yes |
| Joe | 61–65 | 31–35 | 16–20 | Overweight | No | No | No | Yes | Yes |
| Lee | 46–50 | 11–15 | 6–10 | Overweight | Yes | No | No | Yes | No |
| Leo | 76–80 | 36–40 | 1–5 | Normal | Yes | Yes | Yes | Yes | Yes |
| Ned | 56–60 | 16–20 | 16–20 | Overweight | Yes | No | No | Yes | Yes |
| Tim | 36–40 | 6–10 | NR | Overweight | Yes | Yes | No | Yes | Yes |
| Ben | 56–60 | 21–25 | 11–15 | Overweight | Yes | Yes | Yes | No | No |
| Cam | 51–55 | 26–30 | 1–5 | Overweight | Yes | No | No | No | No |
| Fin | 31–35 | 6–10 | 6–10 | Overweight | Yes | No | No | No | No |
| Ken | 56–60 | 26–30 | 6–10 | Overweight | Yes | Yes | No | No | Yes |
| Ric | 66–70 | 16–20 | 1–5 | Obese | Yes | Yes | Yes | No | No |
| Ron | 51–55 | 16–20 | 16–20 | Normal | Yes | Yes | No | No | Yes |
| Sam | 56–60 | 21–25 | 16–20 | Overweight | Yes | Yes | No | No | No |
| Wes | 66–70 | 26–30 | 21–25 | Overweight | Yes | No | No | No | Yes |

Body mass index=categorised with reference to WHO international classification guidelines (normal weight: 18.9–24.9 kg/m², overweight: 25.0–29.9 kg/m², obese: ≥30.0 kg/m²)[51]; joint pain = 'Do you experience pain, discomfort or have a problem with your hip(s) or groin, knee(s), ankle(s), spine (back or neck), shoulder(s), elbow(s), wrist(s), finger(s) or hand(s)?'.
Participants above the horizontal line were meeting the UK Physical Activity Guidelines[32] and participants below the horizontal line were not.
Osteoarthritis = 'Have you ever been told you have wear and tear, degeneration or osteoarthritis by a doctor?'; total joint replacement=have you ever had joint replacement surgery?; UK professional seasons=number of seasons played of professional cricket in the UK.
*Ranges were reported rather than absolute values to assure participants' anonymity.
NR, not reported.

On the other hand, sufficiently active participants had formed strong physical activity habits by integrating physical activity into their daily routine.

Joe: There's not a lot more I could do really, you know, I try and do 10,000 steps a day, I cycle twice a week, I go to the gym a couple of times a week … I'm sort of set in my routine if you will.' *(61–65 years, sufficiently active, satisfied with activity level, no joint pain)*

### Intrinsic and extrinsic motivation

Sources of motivation to undertake physical activity differed between retired cricketers with contrasting activity levels. Cricketers who were not meeting the physical activity guidelines despite expressing dissatisfaction with current activity levels relied on others for motivation to participate in physical activity.

Interviewer: 'Is cricket still a part of your life, today?'

Cam: 'Not really, although I have a six year old son, so it's starting to come back in because I am starting to take him and, and practice with him and coach

him and stuff like that, so but no it hasn't really been part of my life at all for the last 10 years.' *(51–55 years, insufficiently active, dissatisfied with activity level, current joint pain)*

Ric: 'Well I'm ashamed to admit it but not many at the moment, as I said I need to shake myself and get up and get out and do something a bit more and I think my wife will galvanise me and say right we are off for a fast walk for 2 or 3 miles, 2 or 3 times a week to try and sort of get back to what we were doing.' *(66–70 years, insufficiently active, dissatisfied with activity level, current joint pain)*

Participants who were insufficiently active and expressed little or no desire to increase activity levels did not see physical activity as congruent with their current sense of self or identity and were lacking internal motivation to exercise.

Interviewer: 'If you wanted to increase your physical activity levels, what do you think would help you to do so?'

Ron: '… there isn't really anything you know, maybe my kids … as long as my mind is active, physical activity, you know, isn't something that, it's never really jumped out at me … I would say quality of life is pretty good and I don't really have any desire to put on a tracksuit at 52 and become a trendy middle aged man who goes for a jog around the block, like I see many people doing.' *(51–55 years, insufficiently active, satisfied with activity level, current joint pain)*

Ken: 'Um, I don't know a 25 year-old girlfriend who wanted to go cycling. Yeah, I know that sounds flippant; but it's probably true.' *(56–60 years, insufficiently active, satisfied with activity level, current joint pain)*

In contrast, people meeting or exceeding the physical activity guidelines who were satisfied with their current activity level described intrinsic sources of motivation and emphasised the importance of physical activity in maintaining optimal mental and physical well-being across the lifespan.

Leo: 'I do it because I love it. I don't do it because I have to do it, but I am not like some of my friends who say, look I've got to go to walk this morning or I've got to go to the gym and swim for half an hour and I've got to do my weights and all this type of thing, I do it because I love it. I simply love it. If I don't exercise and do the things that I like I get quite, I can actually get quite crotchety and short tempered because I feel frustrated.' *(76–80 years, sufficiently active, satisfied with activity level, current joint pain)*

Ned: 'I can only go from how I feel personally. I mean I feel a lot better doing some form of exercise … you know I think physically you feel better also mentally for the rest of your life, whatever you're doing, you know certainly for me it's a very important part of keeping myself motivated in life as much as anything I guess.' *(56–60 years, sufficiently active, satisfied with activity level, current joint pain)*

### Physical activity preferences
Individuals not meeting the physical activity guidelines commonly expressed experiencing little enjoyment partaking in unaccompanied recreational exercise such as cycling and gym-based activities, with a preference to be active through sport participation. Some insufficiently active individuals described having never enjoyed maintaining fitness or the monotonous aspects of cricket training, but participated reluctantly in order to get out on the field and play cricket which brought them great satisfaction.

Fin: 'The gym side and the physical side of professional sport was the bit that I liked the least … So I was almost rebelling if you like, saying 'haha', I don't have to do this anymore, so I'm not going to. But it was a dreadful decision really. Because it's obviously not very good for you … The monotony of going to the gym and doing half an hour on the treadmill

for example, I can't physically do it. Actually that's wrong I can physically do it, I can't mentally do that. So the type has to be sort of something I enjoy and I guess that's why I do football really, because I enjoy that and it's competitive. I don't find the going to the gym scenario a very appealing one.' *(31–35 years, insufficiently active, dissatisfied with activity level, current joint pain)*

Ken: 'I enjoyed playing the sport; I will admit that I never enjoyed getting fit for it, but it was something you had to do and when it's no longer your living and there is no need to get up at 6 o'clock and go running or doing other, you know fitness exercises or whatever, it was a tremendous relief, shall we say.' *(56–60 years, insufficiently active, satisfied with activity level, current joint pain)*

In contrast, sufficiently active individuals participated in some form of independent recreational activity, such as cycling, running or gym-based exercise. For these individuals, participating in physical activity was more important than the specific type of exercise, and most were willing to sacrifice some enjoyment if physical limitations led them to substitute their favourite form of exercise for a less preferred form of exercise to enable continuation of a physically active lifestyle.

Joe: 'I've always run, I've always' run … but like I say now I can't, I haven't done it for about two years, so I am making do with cycling now. I mean I still get a buzz out of it, but it's not the same as running. I just like, you know, to do something.' *(61–65 years, sufficiently active, satisfied with activity level, no joint pain)*

Ned: 'I'd be happy to do anything I'm capable of doing, but I've kind of got you know my routines now and obviously I vary the aerobic work depending on umm, you know how I feel really.' *(56–60 years, sufficiently active, satisfied with activity level, current joint pain)*

### Pain and physical impairment
Despite most individuals experiencing pain and physical impairment, this did not prevent participants from being physically active. Rather, for some individuals, pain and physical impairment affected the type of activity they chose to take part in and imposed limitations participating in higher-impact activities.

Leo: 'I would like to be able to get out there and run for 40–50 min without any knee problem and pain and going under the knife. But then I am thinking about having one done [a knee replacement] so I can run in marathons or half marathons when I'm over 80 and among that age group.' *(76–80 years, sufficiently active, satisfied with activity level, current joint pain)*

Lee: 'Yeah, yeah I mean I can get by with my knees, but like my hip, my left hip is shot really, so you know if there are certain things I do, I'm hobbling around for a good week afterwards and you know it just stops me sort of doing anything too extreme.' *(46–50 years,*

*sufficiently active, dissatisfied with activity level, current joint pain)*

## Cricket coaching

All six retired cricketers who regularly coached cricket were able to maintain a physically active lifestyle. This in part was due to active involvement in training drills and warm-up sessions. Being around a sporting environment provided motivation to maintain fitness, and coaching cricket provided the time and resources needed to do so.

Ned: 'I think being in a professional environment encourages you to obviously stay fit, you know, you're around professional athletes so you don't want to look fat and incapable of doing your job. So I think that motivates me to keep training … at least with this job I've got time to train, you know, I can do it in the hours that suit me as opposed to having to wait until I finish work at you know 6 o'clock or whatever.' *(56–60 years, sufficiently active, satisfied with activity level, current joint pain)*

Dom: 'I know that my physical activity, I can compensate or counter it by coaching, because I can do more active sessions involving myself if I need to and set standards in that, so I drive other people to do what I think they should be achieving.' *(61–65 years, sufficiently active, satisfied with activity level, no joint pain)*

The positive impact of cricket coaching on physical activity levels was further demonstrated by Sam, who described having been active while he was coaching cricket, but was no longer meeting the physical activity guidelines since he stopped coaching.

Sam: 'I coached there for just under 19 years, so you know I was quite active with the lads there … I used to hit all the catches and do all the fielding drills for the cricket team … it was just like part of my life, when the lads started I'd join in or some days the lads wouldn't be in at all, so I would then make an effort and go to the gym and do stuff and I had my own routines, so yeah, it was quite active really. But I retired … so I haven't, so I've sort of done less exercise.' *(56–60 years, insufficiently active, dissatisfied with activity level, current joint pain)*

## Practical strategies for promoting a physically active lifestyle after retirement

A number of suggestions were made by participants regarding strategies for adopting an active lifestyle after retirement from cricket and other useful information arose from exploring factors influencing physical activity choices. This information guided five recommendations for optimising physical activity across the lifespan after retiring from professional cricket.

## Prioritise physical activity

Retirement from sport may mark a pivotal point in one's life where decisions surrounding physical activity have great potential to impact physical activity levels and health in later life. Prioritising physical activity may be a means to overcome the most commonly acknowledged barrier to being more physically active in this sample of retired cricketers, time constraints.

Leo*:* There is no excuse for people not keeping fit after playing professional cricket, no excuse at all. If you're a married man, kids, things like that, people work long hours these days, how do you squeeze it in? Well you squeeze it in by doing a 25 min run whilst your kids are in the bath, you come back and take them out and dry them and put them to bed and help mum, that type of thing.' *(76–80 years, sufficiently active, satisfied with activity level, current joint pain)*

## Establish a physical activity plan prior to retirement and don't take a break from physical activity

Establishing a physical activity plan prior to retirement and advice to not take a break from physical activity after retirement were suggested by participants as strategies to encourage adoption of routines and habits that are conducive to living a physically active life.

Gus: 'Well, I think you have, you have two choices, you have your own choice and I think it's really that choice of, of getting off your back side and having a plan. You know you must have a plan for your well-being, but you know it's fitness as you get older just doesn't happen, we all think we are invincible when we are 21 and you know, getting on with our lives, but the reality strikes I suppose. You need a plan and I think if you're that way organised, you can go and get some support as well, find a buddy and do it, that's the key.' *(56–60 years, sufficiently active, dissatisfied with activity level, no joint pain)*

Fin: 'The advice I would give from my personal experience is to, to get into the habit of doing something regularly straight away. That is the advice I would give.' *(31–35 years, insufficiently active, dissatisfied with activity level, current joint pain)*

## Evaluate sources of physical activity motivation and incorporate these into a physical activity plan

When establishing a physical activity plan, cricketers may benefit from evaluating sources of physical activity motivation. People who are externally motivated could benefit from tailoring activity choices to satisfy their external sources of motivation (eg, coaching a cricket team, exercising with family or friends, or committing to an exercise group or sports team). Other individuals who are motivated by a desire to compete may be best suited to specific activities that satisfy competitive desires without exacerbating joint pain and function.

Lee: 'I think the big thing for people is finding something that, that clicks with them, that just catches their imagination when they're playing it and so for me, you know, golf is something that does that, surfing is something that is a totally different thing which I learnt after I played cricket … I think that's the crux of it, it's finding something that just keeps you motivated to get out there and enjoy yourself really.' *(46–50 years, sufficiently active, dissatisfied with activity level, current joint pain)*

Ben: 'What advice would I give them? Just to maintain your interest in the game if you can, or some aspect of any game, just to fulfil your sort of competitive instincts if they still remain.' *(56–60 years, insufficiently active, dissatisfied with activity level, current joint pain)*

### Find multiple, satisfying forms of physical activity that can be adapted to accommodate fluctuations in physical capabilities across the lifespan

Physical activity plans should include multiple sources of satisfying physical activity, alternative sources of physical activity may be required if preferred activities become limited due to age, joint pain or physical limitations.

Guy: 'I think you need to try and find something that is linked to that and gives you that same satisfaction and same buzz and that same adrenalin rush, but is mirrored with your body and your age and your lifestyle. And I think trying to find that is not easy, but that's one thing that I have certainly found with surfing, is that I want to try and compete and be good at it and … you don't have to worry about an age thing, it's not necessarily a barrier to being good and competing and so that would be my advice.' *(46–50 years, sufficiently active, dissatisfied with activity level, current joint pain)*

Jim: 'I'm going down this afternoon and I'm quite looking forward to it. I'm going to have to change what I do because my ankles are a bit sore, I've been on the bike perhaps too much pressure on, and I'll have to go on a rowing machine and have a swim. So it's that sort of thing, if I go on the rowing machine too much my back starts to ache, so I've got to go back on the bike.' *(66–70 years, sufficiently active, satisfied with activity level, current joint pain)*

### Coach cricket

Cricketers who are concerned about maintaining an active lifestyle could consider cricket coaching. All coaches in this study were meeting the Physical Activity Guidelines.

Gus: 'Well everyone always gives the reason that, or gives the excuse that there are not many coaching jobs. Well there is perhaps not many coaching jobs at the top end, but there are coaching jobs out there and with the resources that the players have to be able to get qualified as coaches during the period that they are playing, and these courses are paid for,

I mean that's what I did, I got myself qualified in that sense and it allowed me to sort of seamlessly move into a coaching career.' *(56–60 years, sufficiently active, dissatisfied with activity level, no joint pain)*

## DISCUSSION

Retired cricketers' personal perspectives and experiences have enabled identification of key influences on physical activity behaviours. These were (i) time constraints, (ii) habit formation, (iii) intrinsic and extrinsic motivation, (iv) physical activity preferences, (v) pain and physical impairment and (vi) cricket coaching. A number of suggestions were made by participants regarding strategies for adopting an active lifestyle after retirement and other useful information arose from exploring factors influencing physical activity choices. This information guided five recommendations for optimising physical activity across the lifespan after retiring from professional cricket: (i) prioritise physical activity; (ii) establish a physical activity plan prior to retirement and don't take a break from physical activity; (iii) evaluate sources of physical activity motivation and incorporate these into a physical activity plan; (iv) find multiple, satisfying forms of physical activity that can be adapted to accommodate fluctuations in physical capabilities across the lifespan; and (v) coach cricket.

### Key influences on physical activity behaviours

Several factors influencing physical activity choices were not unique to retired professional cricketers. Time constraints have been identified as a barrier to physical activity in other male groups including those living in rural areas,[34] university employees,[35] patients with prostate cancer[36] and African-Americans.[37 38] Additionally, intrinsic forms of motivation have been shown to predict long-term exercise adherence in a variety of samples.[38] A strong preference for competitive sport over recreational exercise was found to be a risk factor for adopting an inactive lifestyle 5–20 years after anterior cruciate ligament (ACL) reconstruction in people with knee difficulties.[39] Although these barriers to physical activity may be applicable to the general population, the characteristics of retired cricketers are different from the general population. The journey from playing youth cricket to retiring from professional cricket exposes an individual to a high volume of physical activity and results in the refinement of physical skill and psychological attributes necessary to perform at an elite level. Another key difference between an elite athlete and the general population is that retirement from professional sport provides a novel opportunity where effort can be directed to optimise the likelihood that a retiring athlete transitions into a physically active lifestyle, and maintains it throughout later life.

On the other hand, contrasts were evident regarding the relationship with joint pain and physical activity in our sample of retired cricketers and previous research in this area. A review of the literature confirms that individuals

with osteoarthritis are less active than those without[40] and osteoarthritis is often perceived by those with the disease as a barrier to physical activity.[41] A proportion of people living with osteoarthritis express a misconception that exercise will exacerbate osteoarthritis symptoms and hold pain-avoidance behaviours which become a barrier to being physically active.[42 43] In contrast, former elite cricketers did not express such beliefs, and osteoarthritis or joint pain did not prevent participants from being physically active. It is possible that exposure to professional cricket desensitised participants to exercising through pain or discomfort. Another contributing factor may be the common attributes that these retired cricketers possess, including resilience, a positive outlook, high quality of life, increased body awareness and an ability to adapt activity choices in line with physical capabilities which may enhance one's ability to be active in the presence of chronic joint pain.[44] This is in line with previous research that identified psychological factors as a stronger determinant of physical activity levels than pain severity in individuals with osteoarthritis and chronic pain.[41 45 46] These findings support further research into the relationship between physical activity, joint pain and quality of life in retired athletes.

### Practical strategies for promoting a physically active lifestyle after retirement

Retiring cricketers could benefit from being informed of the importance of prioritising and maintaining a physically active lifestyle after retirement from cricket. Education may assist with forming intentions to facilitate behaviour change and healthy habit formation.[47] Planning can help to overcome the difficult step of translating intentions into actions, which can lead to habit formation.[48] Making physical activity behaviours habitual has several benefits; forming a habitual physical activity behaviour may reduce the effort required to take part in an activity and promotes continuation of that activity even in times where motivation and self-control are exhausted.[47 49] Specific tools exist which could be used to enable identification of individuals with poor physical activity habits and assess the effectiveness of interventions aimed at facilitating new physical activity habits or changing old habits.[49] Such interventions could draw on habit formation principles such as using repetition, linking activity to consistent cues and performing activity in a similar context to promote automaticity.[47] Changes to an individual's environment or living circumstance (such as retiring from professional cricket and transitioning to postretirement life) provides an opportune time to implement behaviour change and habit formation strategies.[47]

Retiring and former cricketers could also benefit from evaluating what motivates them to be physically active and identifying multiple sources of physical activity tailored to their unique needs and motivations. For individuals largely motivated by competitive team-based environments, this may be of particular importance since some individuals take many years to identify alternative sources of satisfying physical activity after ceasing competitive sport.[39] This has potential to result in adoption of an inactive lifestyle with negative impacts on health and quality of life.[39] Retiring and former cricketers who lack intrinsic motivation to be active could benefit from interventions to foster intrinsic motivation towards physical activity. Such interventions may draw on self-determination theory and cognitive evaluation theory, which emphasise the importance of satisfying an individual's need for competence and autonomy in order to foster intrinsic motivation.[50] Retiring and former cricketers who are externally motived may also benefit from establishing a physical activity plan that incorporates external sources of motivation (such as coaching, team sport or group exercise). All coaches in this study were meeting the Physical Activity Guidelines, yet the positive relationship between cricket coaching and physical activity may be overlooked when this option is considered prior to retiring and transitioning from professional cricket.

### Strengths and potential limitations

Our purposive recruitment strategy may have reduced the generalisability of results since retired cricketers reporting uncertainty regarding the impact of cricket on their physical activity level were not invited into the study. The study may have been subjected to selection bias; individuals who desire participation in a qualitative interview study may share specific attributes that differ from those who decline participation. Notably, six participants reported joint pain without a diagnosis of osteoarthritis and participants spoke about pain and physical impairments as opposed to osteoarthritis in relation to physical activity. For these reasons, we refer to 'pain and physical impairment' rather than osteoarthritis in the results section but draw on osteoarthritis literature to aid with interpretation of findings. We also acknowledge that using self-report to assess physical activity levels and categorise participants into sufficiently active and insufficiently active groups has limitations. Categorising into two groups based on the UK Physical Activity Guidelines resulted in a loss of information presented to the reader regarding highly active and completely inactive individuals. Participants were not contacted after the initial interview for correction or further comment (although they were invited to provide feedback on the manuscript draft); these procedures could have elicited additional insights beyond those gained through the interviews.

This was the first study to explore physical activity in former elite cricketers. The interviewer was a physiotherapist with knowledge of cricket and sports medicine and experience in interviewing and building rapport with individuals. Strong rapport enabled participants to share personal perspectives in a reflective and open manner that enriched the findings of this study.

### CONCLUSION

This study highlights the key influences on physical activity behaviours in retired professional cricketers and provides practical strategies to support retiring and former cricketers to adopt sustainable, physically active lifestyles.

**Acknowledgements** The authors thank the retired cricketers who took part in the interviews. They also acknowledge Angus Porter and the Professional Cricketers' Association for assisting with recruitment and questionnaire development for the larger cross-sectional study from which study participants were purposely recruited.

**Contributors** SRF, FLB, NP and NKA conceived and designed this qualitative study. SRF and MEJ recruited participants and extracted data form the cross-sectional cohort. SRF performed all interviews and drafted the first version of the manuscript. SRF and FLB participated in the analysis. All authors contributed in revising the manuscript and gave their final approval of the submitted version.

**Funding** SRF was awarded a research fellowship from the Arthritis Research UK Centre for Sport, Exercise and Osteoarthritis to support this research. This study received funding from the Arthritis Research UK Centre for Sport, Exercise and Osteoarthritis, as well as the England and Wales Cricket Board.

**Competing interests** NKA and MEJ have received an unrestricted research grant from the England and Wales Cricket Board. NP is employed as the Chief Medical Officer of the England and Wales Cricket Board.

**Ethics approval** Medical Sciences Inter-divisional Research Ethics Committee (IDREC), University of Oxford (reference number R45197/RE001).

**Provenance and peer review** Not commissioned; externally peer reviewed.

**Data sharing statement** To view interview transcripts or additional participant quotes, please contact the corresponding author.

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
