## [Reviewer comments · BMJ Open]

ARTICLE DETAILS

TITLE (PROVISIONAL)	Physical activity in former elite cricketers and strategies for promoting physical activity after retirement from cricket: A qualitative study
AUTHORS	Filbay, Stephanie; Bishop, Felicity; Peirce, Nicholas; Jones, Mary; Arden, Nigel

VERSION 1 – REVIEW

REVIEWER	Miquel Torregrossa Universitat Autònoma de Barcelona Spain
REVIEW RETURNED	09-Jun-2017

GENERAL COMMENTS	The manuscript 'Physical activity in retired professional cricketers and strategies for promoting physical activity after retirement: a qualitative study' is an interesting report of research that can do a valuable contribution to the scientific literature on former professional/elite athletes. However, I have some suggestions trying to help Filbay and colleagues to make their work easier to understand for the readers: P3.L36. 'The participants were living in the United Kingdom or abroad at the time of the interview' it is obvious and it does not add valuable information for the reader. P3. The participants are a part of the method in the abstract and a part of the results in the text. Please place them always in method not in the results section as you do in the manuscript. P3. Report as participants not only the 10 labelled as physically active but also those classified as less active. P7. I think that the reference of the COREQ guidelines is incorrect. It should be Tong, Sainsbury and Graig (2007) instead of Lohamander and colleagues (2007). P10.L96. The questions presented in Table 1 are quite closed and directive it is difficult to accept an inductive thematic approach from those questions. For example, the questions ask about barriers or motivation and then the themes are barriers or motivation. It looks rather a deductive approach than an inductive approach. Please clarify this point. P11.131. As it is in the abstract the participant characteristic is not a result but a part of the method description.
--

	P11. It would help the reader to understand the process the inclusion of a table/figure summarizing the process (including steps from the thematic analysis) from the raw data to the themes. If the authors clarify that it is an inductive approach. P12-16. When possible be consistent presenting always the results from the physically active former professional cricketers and after the physically inactive or vice versa, as it is now it changes from theme to theme. P17.L305. as it is not the first to do so, please add at least one reference of other studies for example: 1. Bäckmand H, Kujala U, Sarna S, Kaprio J. Former Athletes' Health-Related Lifestyle Behaviours and Self-Rated Health in Late Adulthood. Int J Sports Med [Internet]. 2010 Oct 21 [cited 2016 Jul 21];31(10):751–8. Available from: http://www.thieme-connect.de/DOI/DOI?10.1055/s-0030-1255109 2. Bäckmand HM, Kaprio J, Kujala UM, Sarna S. Physical activity, mood and the functioning of daily living A longitudinal study among former elite athletes and referents in middle and old age. Arch Gerontol Geriatr [Internet]. 2009 Jan [cited 2014 Jan 30];48(1):1–9. Available from: http://www.sciencedirect.com/science/article/pii/S0167494307001938
--	---

REVIEWER	Suzanne McDonald Newcastle University, UK
REVIEW RETURNED	18-Jul-2017

GENERAL COMMENTS	Thank you for inviting me to review this article describing the findings from a study exploring influences on physical activity after retirement from professional cricket. Promoting physical activity during life transitions such as retirement represents an important avenue of research for improving public health. I enjoyed reading this paper and have outlined some suggestions/questions for the authors: Abstract  • Can you make it clearer that 'retirement' in this article is retirement from sport and not old-age retirement? I think this is important because old-age retirement will involve different influences on physical activity as shown by existing research in this area • Can you make it clearer why the study focuses on cricketers and not athletes in general? i.e. why is it important to study cricketers specifically? Introduction  • As mentioned above can you make it clearer early on in the introduction that the article is about retirement from sport rather than old-age retirement? Perhaps there needs to be further emphasis about the (younger) age at which retirement from cricket can occur (relative to non-sport occupations)? • There is some inconsistency in the use of terms 'physical activity', 'sport' and 'exercise' (e.g. line 3 and line 6) which could be tightened up in the introduction and other parts of the article e.g. discussion section; line 312 ('physical activity') and line 315 ('exercise'). • It is likely that the social and psychological benefits of cricket also dissipate, in addition to the physical benefits, after retirement (line 7) so perhaps you should mention all three
--

	 • There is some inconsistency in the use of terms which could be tightened up (line 10) – perhaps change from ‘physiological’ to ‘physical’ • Can you clarify and/or provide a reference to support the following sentence: ‘elite athletes who become inactive after retirement face the same, or worse, risk of developing chronic disease...’ (line 9) • Is there any evidence which can help to quantify the size of the target population to give the reader a sense of how relevant/important the study findings are (line 16)? The introduction could be more convincing about the relevance/importance of research in this particular target population • It might be helpful to provide more detail about the training a professional cricketer regularly undertakes during their career (line 19). Later on in the article I learned that training can be intensive and involve gym sessions in between games etc. More information to demonstrate the high frequency/intensity of physical activity involved in cricket may be useful for a naïve reader (like me). • Can you provide references in line 22 • After reading the whole article it seemed to me that a key research question was to compare influences on physical activity across participants according to different characteristics including (a) those meeting the physical activity guidelines/those not meeting the physical activity guidelines (b) those satisfied/those dissatisfied with current physical activity levels. I think the aim should be updated to better reflect this • The wording of the sentence in line 22-26 gives the impression that osteoarthritis is a key research question in this study (and osteoarthritis is discussed in some detail in the discussion). However it isn’t mentioned in the aim and I couldn’t find anything specifically about osteoarthritis in the results section (although there is information about pain and impairment). Perhaps you could highlight differences between those with (n=10) and without (n=8) osteoarthritis in the results if it is a key research question. Methods  • I don’t think there is enough information about the larger cohort study in the method section as it stands. Are there any references available for the cohort study that participants have been recruited from? • Can you provide further details about how you ensured that age was considered in the sampling strategy? • Can you indicate how many individuals from the larger study indicated a willingness to participate? • Check line 67 as it sounds like consent was taken from all participants before the first interview was conducted (perhaps it was). Perhaps dropping ‘18’ would help • How did the pilot testing inform the final interview guide? (line 73-74) • Can you provide a justification for choosing four additional interviews (2 of each activity category) to check for data saturation? A reference would be useful • Can you clarify what you mean by ‘multiple categories’ in line 99 – how did those categories relate to the next stage of data analysis (line 103)? • It is a strength of the study that independent double coding has been conducted. Can you provide an indication of the level of agreement (e.g. %) between the two coders?
--	---

- What reference period was used to categorise participant's physical activity level e.g. physical activity over the last week, month, year? Was the intensity of physical activity considered in accordance with the physical activity guidelines?

- It would be helpful to state how physical activity was defined to participants. Did it include all types of physical activity or did it focus on exercise/sport?

Results

- Line 135 suggests that only frequency (and not intensity) was taken into account when categorising participants according to the UK physical activity guidelines. Was intensity considered too?

- In the results there are some explicit comparisons between participants according to their satisfaction/dissatisfaction with current physical activity but this is not identified as a research question or mentioned in the methods section.

- There is no mention of the practical strategies mentioned by participants in the results section yet these appear in some depth in the discussion section. Perhaps some of the text in the discussion belongs in the results section. A summarised version of this data under 'implications' may work better in the discussion section (see later comments).

- Perhaps after each participant quote you could add further information in brackets e.g. (age range, meets physical activity, satisfied with current level). I appreciate that this information is already provided in a table but I think this would help the reader.

- Within the theme 'intrinsic and extrinsic motivation' presented in line 190 I would like to see whether or not participants who are satisfied vs. dissatisfied with current physical activity had different views, as satisfaction/dissatisfaction with current physical activity may have a strong bi-directional relationship with motivation.

- In line 209 it would be useful to see the interviewer's question

- What do you mean by the 'lowest activity levels' (line 219) – was this all the participants not meeting the physical activity guidelines or was it a sub-set of those participants?

Discussion

- I think it would be useful to follow the journal's suggested discussion headings where possible. These are: "statement of the principal findings; strengths and weaknesses of the study; strengths and weaknesses in relation to other studies, discussing important differences in results; the meaning of the study: possible explanations and implications for clinicians and policymakers; and unanswered questions and future research"

- I would prefer to see a brief summary of the findings in the first paragraph e.g. a list of the key themes under the 'statement of the principal findings'

- The first sentence in the discussion (line 304) might be better placed in the strengths and limitations section.

- The dichotomous cut off used for categorising participant's physical activity is a limitation and so is the use of self-reported physical activity measurement

I believe that all these points can be addressed and I think the manuscript has the potential to make a good contribution to the journal.

VERSION 1 – AUTHOR RESPONSE

Reviewer: 1

Reviewer Name: Miquel Torregrossa

Institution and Country: Universitat Autònoma de Barcelona, Spain

Please state any competing interests or state 'None declared': None declares

Please leave your comments for the authors below

The manuscript 'Physical activity in retired professional cricketers and strategies for promoting physical activity after retirement: a qualitative study' is an interesting report of research that can do a valuable contribution to the scientific literature on former professional/elite athletes. However, I have some suggestions trying to help Filbay and colleagues to make their work easier to understand for the readers:

1.1 Reviewer comment

P3.L36. 'The participants were living in the United Kingdom or abroad at the time of the interview' it is obvious and it does not add valuable information for the reader.

1.1 Author response

Thank you for drawing attention to this. We have removed this statement from the abstract.

1.1 Author action

P3.L36 'All participants formerly played professional cricket in the United Kingdom.'

1.2 Reviewer comment

P3. The participants are a part of the method in the abstract and a part of the results in the text. Please place them always in method not in the results section as you do in the manuscript.

1.2 Author response

In line with the reviewer's recommendation, we have moved the description of participant characteristics to the methods section.

1.2 Author action

Lines 155-164. The participant characteristics paragraph and Table 2 are now presented in the methods section, rather than at the beginning of the results section.

1.3 Reviewer comment

P3. Report as participants not only the 10 labelled as physically active but also those classified as less active.

1.3 Author response

Thank you for this suggestion, we have made the following amendment:

1.3 Author action

Abstract 'Ten participants (56%) were classified as sufficiently active according to the UK Physical Activity Guidelines (moderate intensity activity ≥ 150 minutes per week, or vigorous intensity activity ≥ 75 minutes per week). Eight participants did not meet these guidelines and were classified as insufficiently active.'

1.4 Reviewer comment

P7. I think that the reference of the COREQ guidelines is incorrect. It should be Tong, Sainsbury and Graig (2007) instead of Lohamander and colleagues (2007).

1.4 Author response

Thank you for noting this. While the correct reference was used for this statement, the citations had not been updated through EndNote and consequently the numbers did not accurately correspond with those in the reference list.

1.4 Author action

The citations have now been updated and the reference list is accurate.

1.5 Reviewer comment

P10.L96. The questions presented in Table 1 are quite closed and directive it is difficult to accept an inductive thematic approach from those questions. For example, the questions ask about barriers or motivation and then the themes are barriers or motivation. It looks rather a deductive approach than an inductive approach. Please clarify this point.

1.5 Author response

Although the topic guide addressed broad areas including motivation and barriers, the themes that emerged surrounding these were not specifically asked about and emerged from analysing the interview transcripts in their entirety. For example, although participants were typically asked about their motivation for playing cricket and motivation for taking part in physical activity, the theme related to motivation surrounds intrinsic and extrinsic motivation which was not explicitly inquired about in the topic guide. Rather, these aspects of motivation emerged during analysis and incorporates material raised in response to other interview questions.

This research was exploratory, performed without prior engagement with the literature and without prior hypotheses (lines 124-126). A semi-structured interview guide was used to assure that key areas of interest were covered while enabling the researcher to adapt the interview guide to elicit relevant and rich information from respondents by probing and prompting for additional details. Whilst some questions may seem narrow, the interview guide was tailored to each participant and strategies were employed to facilitate open responses and free speech surrounding broad areas of relevance (see lines 97-102). Attention was paid to commonalities and contrasts between sufficiently active and insufficiently active participants, and the themes presented reflect patterns identified in the data. We hope this response helps clarify the inductive nature of the analysis.

1.5 Author action

No action taken.

1.6 Reviewer comment

P11.131. As it is in the abstract the participant characteristic is not a result but a part of the method description.

1.6 Author response

This was addressed in response 1.2

1.6 Author action

The participant characteristics paragraph and Table 2 are now presented in the methods section, rather than at the beginning of the results section.

1.7 Reviewer comment

P11. It would help the reader to understand the process the inclusion of a table/figure summarizing the process (including steps from the thematic analysis) from the raw data to the themes. If the authors clarify that it is an inductive approach.

1.7 Author response

Thank you for this suggestion, we have included a figure outlining the data analysis process.

1.7 Author action

Figure 1. Data analysis process

1.8 Reviewer comment

P12-16. When possible be consistent presenting always the results from the physically active former professional cricketers and after the physically inactive or vice versa, as it is now it changes from theme to theme.

1.8 Author response

Thank you for drawing attention to this, we have modified the order of results to ensure this is consistent throughout.

1.8 Author action

Results regarding the insufficiently active participants are now presented first throughout the results section.

1.9 Reviewer comment

P17.L305. as it is not the first to do so, please add at least one reference of other studies for example:
1. Bäckmand H, Kujala U, Sarna S, Kaprio J. Former Athletes' Health-Related Lifestyle Behaviours and Self-Rated Health in Late Adulthood. *Int J Sports Med* [Internet]. 2010 Oct 21 [cited 2016 Jul 21];31(10):751–8. Available from: <http://www.thieme-connect.de/DOI/DOI?10.1055/s-0030-1255109>
2. Bäckmand HM, Kaprio J, Kujala UM, Sarna S. Physical activity, mood and the functioning of daily living A longitudinal study among former elite athletes and referents in middle and old age. *Arch Gerontol Geriatr* [Internet]. 2009 Jan [cited 2014 Jan 30];48(1):1–9. Available from: <http://www.sciencedirect.com/science/article/pii/S0167494307001938>

1.9 Author response

Thank you for sharing these references. This sentence has been removed from the beginning of the discussion as recommended by the second reviewer. We now acknowledge in the strengths and potential limitations section that 'This was the first study to explore physical activity in former elite cricketers' and have removed the statement that it was one of the first to do so in retired athletes.

1.9 Author action

Line 554. 'This was the first study to explore physical activity in former elite cricketers.'

Reviewer: 2

Reviewer Name: Suzanne McDonald

Institution and Country: Newcastle University, UK

Please state any competing interests or state 'None declared': None declared

Please leave your comments for the authors below

BMJ open review: Physical activity in retired professional cricketers and strategies for promoting physical activity after retirement: A qualitative study
bmjopen-2017-017785

Thank you for inviting me to review this article describing the findings from a study exploring influences on physical activity after retirement from professional cricket. Promoting physical activity during life transitions such as retirement represents an important avenue of research for improving public health. I enjoyed reading this paper and have outlined some suggestions/questions for the authors:

2.1 Reviewer comment

Abstract

- Can you make it clearer that 'retirement' in this article is retirement from sport and not old age retirement? I think this is important because old-age retirement will involve different influences on physical activity as shown by existing research in this area

2.1 Author response

Thank you for highlighting this. This has now been clarified throughout the abstract and the title has been revised accordingly.

2.1 Author action

Title: 'Physical activity in former elite cricketers and strategies for promoting physical activity after retirement from cricket: A qualitative study'

'The health benefits of professional sport dissipate after retirement unless an active lifestyle is adopted, yet reasons for adopting an active or inactive lifestyle after retirement from sport are poorly understood.'

'...establish a physical activity plan prior to retirement from cricket and don't take a break from physical activity'

'We aimed to identify influences on physical activity behaviours after retirement from professional cricket in sufficiently active and insufficiently active... and provide practical strategies for promoting physical activity after cricket retirement.'

2.2 Reviewer comment

- Can you make it clearer why the study focuses on cricketers and not athletes in general? i.e. why is it important to study cricketers specifically?

2.2 Author response

We have added an additional sentence to the objectives section of the abstract to address this. Unfortunately we were unable to elaborate further due to abstract word restrictions.

2.2 Author action

'Elite cricket is all-encompassing, requiring a high volume of activity and unique physical demands. We aimed to identify influences on physical activity behaviours in active and insufficiently active former-elite cricketers and provide practical strategies for promoting physical activity after cricket retirement.'

2.3 Reviewer comment

Introduction

- As mentioned above can you make it clearer early on in the introduction that the article is about retirement from sport rather than old-age retirement? Perhaps there needs to be further emphasis about the (younger) age at which retirement from cricket can occur (relative to non-sport occupations)?

2.3 Author response

Retirement from sport is very different from occupational retirement regarding influences on physical activity. We have made this distinction more clear throughout the manuscript to assure the reader does not interpret this as occupational retirement. Additionally we have added text to the introduction to further clarify this distinction..

2.3 Author action

Line 29-36: 'The Professional Cricketers' Association (PCA) published an online report from a past player survey of 506 former cricketers aged a mean 49 (range 22 to 86) years.¹ The PCA reported that 88% of former cricketers needed to find work after retiring from cricket and 20% suffered health consequences from playing sport.¹ Transitioning from professional sport to a sedentary profession and health consequences from playing sport have potential to impact physical activity behaviours in former cricketers, although this was not investigated.'

2.4 Reviewer comment

- There is some inconsistency in the use of terms 'physical activity', 'sport' and 'exercise' (e.g. line 3 and line 6) which could be tightened up in the introduction and other parts of the article e.g. discussion section; line 312 ('physical activity') and line 315 ('exercise').

2.4 Author response

Thank you for highlighting this. We have made revisions where appropriate throughout the article to use physical activity as opposed to exercise. However, there are instances where exercise is appropriate for use, such as discussing exercise adherence and recreational exercise.

2.4 Author action

We have made several changes throughout the manuscript from exercise to physical activity, these are highlighted as track changes.

2.5 Reviewer comment

- It is likely that the social and psychological benefits of cricket also dissipate, in addition to the physical benefits, after retirement (line 7) so perhaps you should mention all three

2.5 Author response

Our recent work in retired cricketers published in BMJ Open found that former cricketers may continue to benefit, both psychologically and socially, from their past career in cricket. We have discussed these study findings in the discussion section. In light of these findings, we would prefer not to alter this sentence.

2.5 Author action

No changes.

2.6 Reviewer comment

- There is some inconsistency in the use of terms which could be tightened up (line 10) – perhaps change from ‘physiological’ to ‘physical’

2.6 Author response

Thank you for this suggestion, we have replaced ‘physiological’ with ‘physical’

2.6 Author action

Line 10. ‘If the physical and psychological benefits an athlete obtained through professional sport could be maintained by adopting a physically active lifestyle after retirement..’

2.7 Reviewer comment

- Can you clarify and/or provide a reference to support the following sentence: ‘elite athletes who become inactive after retirement face the same, or worse, risk of developing chronic disease....’ (line 9)

2.7 Author response

While the correct references were used within the text, the citations had not been re-updated through EndNote prior to submission and consequently the numbers did not correctly correspond with those in the reference list.

2.7 Author action

The citations have now been updated and the reference for this statement is now accurate.

2.8 Reviewer comment

- Is there any evidence which can help to quantify the size of the target population to give the reader a sense of how relevant/important the study findings are (line 16)? The introduction could be more convincing about the relevance/importance of research in this particular target population

2.8 Author response

We have added additional information from the Professional Cricketers’ Association (PCA) past player survey of 506 former cricketers to the introduction. We believe this further highlights the relevance/importance of research in former professional cricketers.

2.8 Author action

Line 29-36: ‘The Professional Cricketers’ Association (PCA) published an online report from a past player survey of 506 former cricketers aged a mean 49 (range 22 to 86) years.¹ The PCA reported that 88% of former cricketers needed to find work after retiring from cricket and 20% suffered health consequences from playing sport.¹ Transitioning from professional sport to a sedentary profession and health consequences from playing sport have potential to impact physical activity behaviours in former cricketers, although this was not investigated.’

2.9 Reviewer comment

- It might be helpful to provide more detail about the training a professional cricketer regularly undertakes during their career (line 19). Later on in the article I learned that training can be intensive and involve gym sessions in between games etc. More information to demonstrate the high frequency/intensity of physical activity involved in cricket may be useful for a naïve reader (like me).

2.9 Author response

In line with your suggestion, we have added additional information to the introduction.

2.9 Author action

Lines 19-22: 'During the course of the seven month summer season the playing schedule is relentless and many elite cricketers also play overseas during the winter period. Elite cricketers train during the preseason months and in between games with a mixture of skills practice, aerobic and strength based conditioning.'

2.10 Reviewer comment

- Can you provide references in line 22

2.10 Author response

This statement resulted from a review of the literature and has over 30 accompanying references. For this reason, and since the vast majority of research within cricket surrounds incidence, prevention, prediction and treatment of cricket injuries, we decided to modify the statement to reflect this. We have referenced 8 key studies in this area to support this statement.

2.10 Author action

Line 24. 'The vast majority of cricket-related research has focused on incidence, prevention, prediction and treatment of cricket injuries.2-9'

2.11 Reviewer comment

- After reading the whole article it seemed to me that a key research question was to compare influences on physical activity across participants according to different characteristics including (a) those meeting the physical activity guidelines/those not meeting the physical activity guidelines (b) those satisfied/those dissatisfied with current physical activity levels. I think the aim should be updated to better reflect this

2.11 Author response

Encompassed within the first study aim, was the exploration of patterns and comparison of themes between sufficiently active and insufficiently active participants, and we used the physical activity guidelines to categorise individuals into sufficiently active and insufficiently active groups. Although this was addressed in aim i) ('to draw upon retired-cricketers' personal perspectives and experiences to identify key influences on physical activity behaviours after retirement from professional cricket') we can see how this could be better reflected with a slight modification to the wording of this aim.

2.11 Author action

The wording of the first aim has been updated in the abstract and introduction to better reflect the exploration of themes between sufficiently active and insufficiently active individuals:

- i) identify key influences on physical activity behaviours after retirement from professional cricket in sufficiently active and insufficiently active individuals

2.12 Reviewer comment

- The wording of the sentence in line 22-26 gives the impression that osteoarthritis is a key research question in this study (and osteoarthritis is discussed in some detail in the discussion). However it isn't mentioned in the aim and I couldn't find anything specifically about osteoarthritis in the results section (although there is information about pain and impairment). Perhaps you could highlight differences between those with (n=10) and without (n=8) osteoarthritis in the results if it is a key research question.

2.12 Author response

We have made some amendments to explain why osteoarthritis was not specifically reported upon in the results despite it being of relevance for the introduction and discussion sections. As shown in Table 2, there was discordance between joint pain and osteoarthritis amongst participants. This is in line with current evidence highlighting a mismatch between knee pain and osteoarthritis (Table 2). Additionally, pain and physical limitations was what was discussed by participants as opposed to 'osteoarthritis,' for these reasons, we presented 'pain and physical impairment' in the results as opposed to osteoarthritis. Had osteoarthritis been identified as a key barrier to physical activity by participants, rather than pain and physical impairment, this would have been reported in the results, however this was not the case.

Although we have chosen not to highlight differences between those with and without osteoarthritis in the results section as suggested for the aforementioned reasons, we did make amendments to the text and limitations section to provide further clarity.

2.12 Author action

Line 27-29: 'Developing osteoarthritis or chronic pain after retirement from cricket has potential to negatively impact physical activity levels in later life, although this has not yet been explored.'
Was modified to.. 'Developing symptomatic osteoarthritis after retirement from cricket has potential to...'

We added an additional sentence to the participant characteristics section to describe the prevalence of osteoarthritis and joint pain in this sample:

Lines 543-549: 'Notably, six participants reported joint pain without a diagnosis of osteoarthritis and participants spoke about pain and physical impairments as opposed to osteoarthritis in relation to physical activity. For these reasons, we refer to 'pain and physical impairment' rather than osteoarthritis in the results section but draw upon osteoarthritis literature to aid with interpretation of findings.'

2.13 Reviewer comment

Methods

- I don't think there is enough information about the larger cohort study in the method section as it stands. Are there any references available for the cohort study that participants have been recruited from?

2.13 Author response

The first manuscript from the larger cohort study is currently under-review, so unfortunately it cannot be referenced. We have however, included reference to a peer-reviewed abstract published from the larger cohort study which provides additional information. We have also provided some additional information in the methods section.

2.13 Author action

A reference was added to provide further information regarding the larger cross sectional study from which participants were recruited.

Line 53-61. Participants were purposively sampled from a cohort of 187 former elite English cricketers. The cohort had been recruited from the former player membership list maintained by the PCA as part of a cross-sectional retrospective questionnaire study. The questionnaire collected information regarding cricket playing history, injury history, current joint health, medical history and demographics.²⁵ From this larger cohort, 143 participants indicated a willingness to be contacted again and formed the cohort from which participants were invited to the current study.

2.14 Reviewer comment

- Can you provide further details about how you ensured that age was considered in the sampling strategy?

2.14 Author response

We have added more information to the recruitment section of the methods.

2.14 Author action

Line 59-62: 'When selecting former cricketers to invite into the study, potential participants were purposely selected to ensure the sample represented men of varying ages. When an individual declined the invitation, a former cricketer of a similar age was invited into the study.'

2.15 Reviewer comment

- Can you indicate how many individuals from the larger study indicated a willingness to participate?

2.15 Author response

We have added that 143 participants had indicated willingness to be contacted again.

2.15 Author action

Line 63-65: 'From this larger cohort, 143 participants indicated a willingness to be contacted again and formed the cohort from which participants were invited to the current study.'

2.16 Reviewer comment

- Check line 67 as it sounds like consent was taken from all participants before the first interview was conducted (perhaps it was). Perhaps dropping '18' would help

2.16 Author response

Informed verbal consent was obtained at the beginning of each interview. We agree that this could be clearer, and have modified the sentence as follows:

2.16 Author action

'line 77-78. 'Informed verbal consent was obtained from each of the 18 participants prior to performing audio recorded semi-structured telephone interviews'

2.17 Reviewer comment

- How did the pilot testing inform the final interview guide? (line 73-74)

2.17 Author response

We have added this information to the methods section.

2.17 Author action

Lines 84-86: The semi-structured interview guide was pilot tested with three people cricket participants experience prior to ethics approval. 'This resulted in the addition of three questions (Q2, Q4, Q15) and the modification of one question (Q17) to assess the perceived relationship between current quality of life and an individual's past cricket career (Table 1).'

2.18 Reviewer comment

- Can you provide a justification for choosing four additional interviews (2 of each activity category) to check for data saturation? A reference would be useful

2.18 Author response

The primary reason for conducting additional interview was to expand upon ideas and themes. Since this was performed after reaching data saturation, we had to have a prior plan in case new themes emerged during these additional interviews. It was decided that additional interviews would be performed until data saturation was again satisfied. However, this was not necessary. Note, the primary purpose of these additional interviews were not to check for data saturation, rather, this was achieved in the process. The wording has been modified to better reflect this by removing the words 'affirm data saturation' from the first sentence below:

2.18 Author action

Line 101-102: 'Once data saturation was reached, an additional four interviews were performed to expand upon ideas and themes after following the semi-structured interview guide. If these final interviews resulted in the identification of new themes, additional interviews were planned until data saturation was again satisfied.'

2.19 Reviewer comment

- Can you clarify what you mean by 'multiple categories' in line 99 – how did those categories relate to the next stage of data analysis (line 103)?

2.19 Author response

In line with the other reviewers' suggestion, we have included a figure outlining how these stages of analysis fit together and inter-relate.

2.19 Author action

Figure 1. Data analysis process

2.20 Reviewer comment

- It is a strength of the study that independent double coding has been conducted. Can you provide an indication of the level of agreement (e.g. %) between the two coders?

2.20 Author response

We cannot provide an exact percentage of agreement, since the second investigator was blinded to the coding structure developed by the first author. However, full agreement was achieved regarding key themes related to these 6 transcripts after discussion between authors and no modifications were made to the first authors' coding structure. The text has been modified to highlight this.

2.20 Author action

Lines 139-142: A meeting was then held between investigators and agreement was achieved regarding key themes in relation to these transcripts. Although no modifications were made to the coding structure following this meeting, the second investigator contributed to the consolidation and interpretation of key themes.

2.21 Reviewer comment

- What reference period was used to categorise participant's physical activity level e.g. physical activity over the last week, month, year? Was the intensity of physical activity considered in accordance with the physical activity guidelines?

2.21 Author response

We have added information to the methods to describe the reference period used to categorise participants' physical activity levels.

The intensity of physical activity was considered when determining whether participants were meeting the UK physical activity guidelines. See line 140-143: 'Physical activity type was categorised into moderate or vigorous intensity with reference to previous recommendations in accordance with Centers for Disease Control (CDC) and American College of Sports Medicine (ACSM) guidelines.¹⁰

2.21 Author action

Line 136-140: 'To enable comparison of physical activity behaviors and perspectives in active and less active counterparts, participant descriptions of current activity level over a typical week were used to categorise participants into 'sufficiently active' (meeting the UK Physical Activity Guidelines¹¹) and 'insufficiently active' (not meeting the UK Physical Activity Guidelines¹¹) groups. Participants were asked to describe any 'physical activity, exercise or sport' that they currently take part in, and were prompted to provide details regarding activity type, duration, intensity and frequency and to assure responses reflected a typical week.'

2.22 Reviewer comment

- It would be helpful to state how physical activity was defined to participants. Did it include all types of physical activity or did it focus on exercise/sport?

2.22 Author response

Participants were asked to describe any 'physical activity, exercise or sport' that they currently take part in.

2.22 Author action

Line 137-140: 'Participants were asked to describe any 'physical activity, exercise or sport' that they currently take part in and were prompted to provide details regarding activity type, duration, intensity and frequency and to assure responses reflected a typical week.'

2.23 Reviewer comment

Results

- Line 135 suggests that only frequency (and not intensity) was taken into account when categorising participants according to the UK physical activity guidelines. Was intensity considered too?

2.23 Author response

Yes, as previously described, intensity was considered. We can see how describing this as 'activity type' may make this unclear, so have modified this to read 'activity type, duration, intensity and frequency'

2.23 Author action

Line 137-140: 'Participants were asked to describe any 'physical activity, exercise or sport' that they currently take part in and were prompted to provide details regarding activity type, duration, intensity and frequency and to assure responses reflected a typical week.'

2.24 Reviewer comment

- In the results there are some explicit comparisons between participants according to their satisfaction/dissatisfaction with current physical activity but this is not identified as a research question or mentioned in the methods section.

2.24 Author response

This was something that was taken into consideration when analysing patterns within the data. The methods has now been amended to reflect this. Note that this information is presented in Table 2 and reported in the participants characteristics paragraph ('One in two (n=9, 50%) would prefer to be participating in a greater volume of physical activity')

2.24 Author action

Lines: 152-157: Key themes and strategies for promoting physical activity will be described with reference to participant quotes^{12 13} and in relation to relevant participant characteristics (i.e. physical activity level, satisfaction/dissatisfaction with activity level and the presence/absence of joint pain).

2.25 Reviewer comment

- There is no mention of the practical strategies mentioned by participants in the results section yet these appear in some depth in the discussion section. Perhaps some of the text in the discussion belongs in the results section. A summarised version of this data under 'implications' may work better in the discussion section (see later comments).

2.25 Author response

We discussed in depth whether these practical strategies would be better presented in the results or the discussion. On further reflection, and in light of your feedback, we have moved the strategies to the results section (with a short description of each strategy and accompanying quotes). The text elaborating upon these strategies in relation to the literature remains in the discussion, with some small revisions to improve flow and formatting.

2.25 Author action

See revisions to the results section under the subheading: 'Practical strategies for promoting a physically active lifestyle after retirement.' This includes the addition of supporting quotes.

Additionally, two paragraphs have been revised in the discussion under the subheading 'Practical strategies for promoting a physically active lifestyle after retirement.'

These revisions are marked with track changes.

2.26 Reviewer comment

- Perhaps after each participant quote you could add further information in brackets e.g. (age range, meets physical activity, satisfied with current level). I appreciate that this information is already provided in a table but I think this would help the reader.

2.26 Author response

Thank you for this suggestion. We have now added participant characteristics below each quote.

2.26 Author action

All quotes in the results section are now presented with participant characteristics. For example:

Fin: 'The advice I would give from my personal experience is to, to get into the habit of doing something regularly straight away. That is the advice I would give.'

(31-35 years old, insufficiently active, dissatisfied with activity level, current joint pain)

The analysis section of the methods has also been updated to reflect this change:

Line 129-131: Key themes and strategies for promoting physical activity will be described with reference to participant quotes^{12 13} and in relation to relevant participant characteristics (i.e. physical activity level, satisfaction/dissatisfaction with activity level and the presence/absence of joint pain).

2.27 Reviewer comment

- Within the theme 'intrinsic and extrinsic motivation' presented in line 190 I would like to see whether or not participants who are satisfied vs. dissatisfied with current physical activity had different views, as satisfaction/dissatisfaction with current physical activity may have a strong bi-directional relationship with motivation.

2.27 Author response

The connection between satisfaction/dissatisfaction and motivation was reported for two of the three paragraphs in this section (i.e. 'Cricketers who were not meeting the physical activity guidelines despite expressing dissatisfaction with current activity levels, relied on others for motivation to participate in physical activity' and 'Participants with the lowest activity levels were lacking internal motivation to exercise, did not see physical activity as congruent with their current sense of self or identity, and expressed little or no desire to become physically active'). We have revised the third paragraph so it also refers to satisfaction with activity level.

2.27 Author action

Additional information regarding satisfaction with activity level was added to this sentence:

Line 244-247: 'In contrast, people meeting or exceeding the physical activity guidelines who were satisfied with their current activity level, described intrinsic sources of motivation and emphasised the importance of physical activity in maintaining optimal mental and physical wellbeing across the lifespan.'

2.28 Reviewer comment

- In line 209 it would be useful to see the interviewer's question

2.28 Author response

The question preceding this quote was: 'Is cricket still a part of your life today? This has been added to the text as suggested.

2.28 Author action

Line 229: Interviewer: 'Is cricket still a part of your life, today?'

2.29 Reviewer comment

- What do you mean by the 'lowest activity levels' (line 219) – was this all the participants not meeting the physical activity guidelines or was it a sub-set of those participants?

2.29 Author response

This paragraph has been reformatted to improve clarity.

2.29 Author action

Line 230-232: Participants who were insufficiently active and expressed little or no desire to increase activity levels did not see physical activity as congruent with their current sense of self or identity and were lacking internal motivation to exercise.

2.30 Reviewer comment

Discussion

- I think it would be useful to follow the journal's suggested discussion headings where possible. These are: "statement of the principal findings; strengths and weaknesses of the study; strengths and weaknesses in relation to other studies, discussing important differences in results; the meaning of the study: possible explanations and implications for clinicians and policymakers; and unanswered questions and future research"

Thank you for this suggestion. We would prefer to leave the formatting as is, having incorporated your previous recommendation to move reporting of strategies to the results section and to summarise the findings in the first paragraph of the discussion. We believe this improves the readability of the discussion and with consideration to the size of the manuscript we would prefer not to restructure the discussion further.

2.30 Author response

No action taken.

2.31 Reviewer comment

- I would prefer to see a brief summary of the findings in the first paragraph e.g. a list of the key themes under the 'statement of the principal findings'

2.31 Author response

Thank you for this recommendation. Paragraph one of the discussion has now been revised to summarise the key findings.

2.31 Author action

Discussion, paragraph one:

Retired cricketers' personal perspectives and experiences have enabled identification of key influences on physical activity behaviours. These were i) time constraints; ii) habit formation; iii) intrinsic and extrinsic motivation; iv) physical activity preferences; v) pain and physical impairment; and vi) cricket coaching. A number of suggestions were made by participants regarding strategies for adopting an active lifestyle after retirement and other useful information arose from exploring factors influencing physical activity choices. This information guided five recommendations for optimising physical activity across the lifespan after retiring from professional cricket: i) prioritise physical activity; ii) establish a physical activity plan prior to retirement and don't take a break from physical activity; iii) evaluate sources of physical activity motivation and incorporate these into a physical activity plan; iv) find multiple, satisfying forms of physical activity that can be adapted to accommodate fluctuations in physical capabilities across the lifespan; v) coach cricket.

2.32 Reviewer comment

- The first sentence in the discussion (line 304) might be better placed in the strengths and limitations section.

2.32 Author response

This sentence has now been moved to the strengths and limitations section as suggested.

2.32 Author action

Line 525: 'This was the first study to explore physical activity in former elite cricketers.'

2.33 Reviewer comment

- The dichotomous cut off used for categorising participant's physical activity is a limitation and so is the use of self-reported physical activity measurement

2.32 Author response

Thank you for highlighting this, we agree, and have acknowledged this in the limitations section.

2.32 Author action

Lines 536-541: 'We also acknowledge that using self-report to assess physical activity levels and categorise participants into sufficiently active and insufficiently active groups has limitations. Categorising into two groups based on the UK Physical Activity Guidelines resulted in a loss of information presented to the reader regarding highly active, and completely inactive individuals.'

On further reflection, using the label 'physically inactive' may be misleading, since this category included individuals participating in physical activity but not to the degree considered to be sufficient according to the UK Physical Activity Guidelines. For this reason, we have changed the terminology throughout the manuscript to refer to those meeting the UK Physical Activity Guidelines at 'sufficiently active' and those not meeting these guidelines as 'insufficiently active.'

2.34 Reviewer comment

I believe that all these points can be addressed and I think the manuscript has the potential to make a good contribution to the journal.

1. The Professional Cricketers' Association (PCA). Past Player Survey 2013 [ONLINE]:Available at: <http://www.thepca.co.uk/assets/files/pdfs/Personal%20Development/Past%20player%20survey%20presentation2013.pdf>. [Accessed 26 July 17].
2. Orchard JW. Injury surveillance in cricket. *British Journal of Sports Medicine* 2013;47(10):605-06.
3. Frost WL, Chalmers DJ. Injury in elite New Zealand cricketers 2002-2008: Descriptive epidemiology. *British Journal of Sports Medicine* 2014;48(12):1002-07.
4. Ranson C, Hurley R, Rugless L, et al. International cricket injury surveillance: A report of five teams competing in the ICC Cricket World Cup 2011. *British Journal of Sports Medicine* 2013;47(10):637-43.
5. Finch CF, Elliott BC, McGrath AC. Measures to prevent cricket injuries. An overview. *Sports Medicine* 1999;28(4):263-72.
6. Morton S, Barton CJ, Rice S, et al. Risk factors and successful interventions for cricket-related low back pain: A systematic review. *British Journal of Sports Medicine* 2014;48(8):685-91.
7. Olivier B, Stewart AV, Olorunju SAS, et al. Static and dynamic balance ability, lumbo-pelvic movement control and injury incidence in cricket pace bowlers. *Journal of Science and Medicine in Sport* 2015;18(1):19-25.
8. Gray J, Aginsky KD, Derman W, et al. Symmetry, not asymmetry, of abdominal muscle morphology is associated with low back pain in cricket fast bowlers. *Journal of Science and Medicine in Sport* 2015.
9. Olivier B, Taljaard T, Burger E, et al. Which Extrinsic and Intrinsic Factors are Associated with Non-Contact Injuries in Adult Cricket Fast Bowlers? *Sports Medicine* 2016;46(1):79-101.
10. Ainsworth BE, Haskell WL, Leon AS, et al. Compendium of physical activities: classification of energy costs of human physical activities. *Med Sci Sports Exerc* 1993;25(1):71-80.
11. Bull FatEWG. Physical Activity Guidelines in the UK: Review and recommendations School of Sport, Exercise and Health Sciences, Loughborough University, May 2010.
12. Braun V, Clarke V. Using thematic analysis in psychology. *Qualitative Research in Psychology* 2006;3(2):77-101.
13. Patton MQ. *Qualitative evaluation and research methods*, 1990.

VERSION 2 – REVIEW

REVIEWER	Dr Suzanne McDonald Institute of Health & Society, Newcastle University UK
REVIEW RETURNED	17-Sep-2017

GENERAL COMMENTS	The author's have done an excellent job in revising the manuscript to address the reviewer comments. I have one outstanding minor query that I would like the authors to address: I now understand that the four additional interviews that were undertaken were not conducted to check for data saturation (thank you for clarifying). However, the reader still does not know what the a priori rule was for checking data saturation in the initial interviews (i.e. those before the 4 additional). Some studies have used an initial sample and a 'stopping criterion' (e.g. see https://www.ncbi.nlm.nih.gov/pubmed/20204937). Can you add a sentence to say how you defined data saturation in your study and at what stage it was achieved?
--

VERSION 2 – AUTHOR RESPONSE

Thank you for taking the time to review the amended version of the manuscript. Please note that we had included this information in the previous version of the manuscript. This was described on page 11 lines 101-108, as follows:

'Data saturation was achieved by the 14th interview, defined a priori as the point at which no new themes were identified from four consecutive interviews (two from participants with increased physical activity and two from participants with decreased physical activity). Once data saturation was reached, an additional four interviews were performed to expand upon ideas and themes after following the semi-structured interview guide. If these final interviews resulted in the identification of new themes, additional interviews were planned until data saturation was again satisfied. No new themes emerged from these additional four interviews affirming data saturation. Data from all 18 interviews were used for analysis.'